# Muscle thickness and inflammation during a 50km ultramarathon in recreational runners

**Rian Q. Landers-Ramos**[1]*, **Kathleen Dondero**[1,2], **Christa Nelson**[3], **Sushant M. Ranadive**[4], **Steven J. Prior**[4,5,6], **Odessa Addison**[2,5]

**1** Department of Kinesiology, Towson University, Towson, Maryland, United States of America, **2** Department of Physical Therapy and Rehabilitation Science, University of Maryland School of Medicine, Baltimore, Maryland, United States of America, **3** Department of Physical Therapy and Human Movement Sciences, Northwestern University Feinberg School of Medicine, Chicago, Illinois, United States of America, **4** University of Maryland, College Park, Maryland, United States of America, **5** Veterans Affairs Medical Center, Geriatric Research and Clinical Center, Baltimore, Maryland, United States of America, **6** Department of Medicine, University of Maryland School of Medicine, Division of Geriatrics and Palliative Medicine, Baltimore, Maryland, United States of America

* rlandersramos@towson.edu

## Abstract

### Purpose

This study examined changes in circulating levels of inflammatory cytokines [IL-6, sIL-6R, TNF-α, and calprotectin], skeletal muscle morphology, and muscle strength following a 50km race in non-elite athletes.

### Methods

Eleven individuals (8 men; 3 women) underwent pre-race assessments of rectus femoris muscle thickness (resting and contracted) using ultrasound, isometric knee extensor torque, and plasma cytokines. Measures were repeated after 10km of running, the 50km finish (post-race), and again 24-hrs post-race.

### Results

Compared with baseline values, Δ muscle thickness (resting to contracted) increased significantly 24 hrs post-race (11 ± 11% vs. 22 ± 8%; P = 0.01). Knee extensor torque was significantly reduced immediately post-race (151 ± 46 vs. 134 ± 43 Nm; P = 0.047) but remained similar to post-race values at 24 hrs post-race (P = 0.613). Compared with pre-race levels, IL-6 and calprotectin concentrations increased 302% and 50% after 10km, respectively (P<0.017 for both), peaked post-race (2598% vs. pre-race for IL-6 and 68% vs. pre-race for calprotectin; P = 0.018 for both), and returned to pre-race levels at 24-hrs post-race (P>0.05 for both). Creatine kinase levels rose steadily during and after the race, peaking 24-hrs post-race (184 ± 113 U/L pre-race vs. 1508 ± 1815 U/L 24-hrs post-race; P = 0.005).

### Conclusion

This is the first report of delayed increases in Δ muscle thickness at 24 hrs post-50km, which are preceded by reductions in knee extensor torque and elevations in plasma IL-6,

**Data Availability Statement:** Data collection for this study took place at a sanctioned 50-kilometer race event in which race results are publicly available. To comply with research ethics and

protect the identity of our participants, we have made a limited data set available to https://doi.org/10.7910/DVN/B6YPF0 that excludes race finish time, age, and sex. All other data have been made available.

**Funding:** The author(s) received no specific funding for this work.

**Competing interests:** The authors have declared that no competing interests exist.

**Abbreviations:** ANOVA, analysis of variance; ASIS, anterior superior iliac spine; BMI, body mass index; CK, creatine kinase; CT, computerized tomography; ELISA, enzyme-linked immunosorbent assay; Km, kilometer; ICC, intraclass correlation of coefficient; IL, interleukin; TNF-α, tumor necrosis factor alpha; MVIC, maximal voluntary isometric contraction; MRI, magnetic resonance imaging; MT, muscle thickness; N, Newton; SD, standard deviation; $VO_{2max}$, maximal oxygen consumption.

and calprotectin. Recreational athletes should consider the acute muscle inflammatory response when determining training and recovery strategies for 50km participation.

## Introduction

Ultramarathon running, broadly defined as any running distance exceeding a traditional 42.2 km marathon, has seen an exponential increase in participation over the past decade [1,2]. Indeed, between 2010–2017, the total participation in ultramarathon events increased nearly 700% [1]. While aerobic exercise generally has a positive impact on cardiovascular outcomes, prolonged aerobic exercise, such as ultramarathon running, can elicit an inflammatory response that may negatively affect muscular function. Ultramarathon running, specifically, has been found to elicit large increases in IL-6 [3–5]. Among its numerous physiological roles, IL-6 may limit motor activity as a protective mechanism in response to fatigue from prolonged or intense exercise [6,7]. Further, when bound to its soluble receptor (sIL-6R), IL-6 acts to promote cellular apoptosis [8] and the inflammatory properties may be amplified [7]. Other inflammatory factors such as tumor necrosis factor (TNF)-α [3,9–11], and calprotectin [12,13] are elevated after prolonged running and are involved in the activation and breakdown of damaged cells, including skeletal muscle cells, through inflammatory signaling pathways [12,14]. Together, these factors may contribute to declines in muscle function (e.g., acute decreases in strength) [15] seen after an ultramarathon event and play a role in subsequent recovery [16]. Swelling or inflammation resulting from an ultramarathon can be assessed as changes in resting muscle thickness (MT) using ultrasound [17,18]. Further, inclusion of MT measures during muscle contraction may provide more insight into changes in muscle morphology post-exercise. For instance, contracted MT may emphasize intracellular swelling due to ions released from cytoskeletal breakdown after repetitive muscle contractions and resulting in shifts in osmotic pressure gradients [19]. Further, contracted MT may reflect altered muscle recruitment (failure to fully recruit or over-recruitment in response to a standard task). These measure can be especially informative when used in conjunction with measures of muscle strength [20]. Therefore, reporting MT under both resting and contracted conditions as well as Δ MT to demonstrate changes relative to resting conditions provides a more comprehensive picture of acute changes in muscle morphology in response to ultramarathon running.

Despite the growing popularity of ultramarathon events, many studies investigating muscular and inflammatory responses thus far have focused on elite athletes in longer duration races. The acute inflammatory and muscular response to prolonged exercise and the recovery timeline may differ between recreationally trained athletes and elite runners. For example, Mooren et al. [13] found that serum calprotectin levels were 1.5-fold higher post-marathon in men with a higher $VO_{2max}$ (>60 ml/kg/min) compared to those with a $VO_{2max}$ < 55 ml/kg/min. Furthermore, though the vast majority of ultramarathon research has been done in races of more extreme ultramarathon distances (100 miles or more) [3,11,21,22], ultramarathon participation levels are highest in 50km distances [1,2]. This points to a gap in knowledge regarding the effects of ultramarathon running on recreational runners participating in 50km events. Understanding the trajectory of the inflammatory process in non-elite athletes and its relationship to skeletal muscle morphological changes and strength during and after these events may improve training and recovery recommendations for average individuals participating in ultrarunning events.

Towards this purpose, we performed a field-based study to examine changes in circulating levels of inflammatory cytokines [IL-6, soluble IL-6 receptor (sIL-6R), TNF-α, and calprotectin], skeletal muscle morphology, and muscle strength during and following a 50km race in a group of non-elite athletes. We hypothesized that resting and contracted MT, as assessed with ultrasound imaging, would increase following a 50km race and this would be accompanied by a decrease in muscle strength. Additionally, we hypothesized that there would be an acute increase in all inflammatory cytokines during and immediately following the 50k race.

## Materials and methods

### Participant selection

Runners planning to participate in a 50-km race in February 2019 were recruited via an email advertisement. Individuals who expressed interest were then screened via telephone call. Inclusion criteria included $\geq$ 18 years old, non-smoker, and a body mass index (BMI) between 18.5–30 $kg/m^2$. Individuals were excluded if they were a recent smoker ($<$ 6 months from smoking cessation), or if they self-reported any medical conditions in which maximal exercise was contraindicated.

### Ethics approval

Participants who met the inclusion criteria provided written informed consent. All procedures performed in studies involving human participants were in accordance with the ethical standards of the institutional and/or national research committee and with the 1964 Helsinki Declaration and its later amendments or comparable ethical standards. The study was approved by the Bioethics Committee at the University of Maryland College Park (IRB #1300927–5).

### Baseline testing

All baseline testing was completed in a single visit ~1–2 weeks prior to the race day in our lab at the University of Maryland. Upon arrival, participants provided informed consent followed by questionnaires documenting previous exercise habits and ultramarathon experience. Seated resting blood pressure was measured with a standard sphygmomanometer on the brachial artery of the participant's dominant arm. Baseline testing included anthropometric assessments, knee extensor maximal isometric strength, muscle ultrasound, and a maximal oxygen uptake test. Details of each of these tests are described below.

### Anthropometric assessments

Body height (m) and mass (kg) were measured using an electronic scale (Seca 264, Hamburg, Germany). Body composition was assessed using air-displacement plethysmography (BodPod, COSMED, Italy).

### Knee extension isometric strength

Maximal voluntary isometric contraction (MVIC) strength of the knee extensors was assessed using handheld MicroFET2 Wireless dynamometer (Hoggan scientific LLC, Salt Lake City, Utah). All knee extensor tests were performed by a single investigator experienced with clinical dynamometry. Participants were seated at the edge of a plinth with 90° knee flexion; measurements of lower leg length were taken from the lateral epicondyle of the femur to the lateral malleolus of the ankle. After taking measurements of both legs, the dynamometer was placed on the anterior leg, level with the lateral malleolus, for each participant. The handheld dynamometer was held in place using a gait belt strapped to the leg of the table to control for test

administrator resistance that may influence test reliability. The location of the dynamometer was marked on the leg using a marker to maintain consistency between trial measurements. Following a familiarization trial, the participants were asked to kick and extend the lower leg out as hard as they could against the dynamometer for approximately 3–5 seconds until a peak isometric force was observed. Knee extensor maximal force was measured in kg and converted to Newtons (N). Knee extension joint torque (Nm) was calculated as the peak dynamometer force in N multiplied by the moment arm of force application (distance from dynamometer to later epicondyle). Each participant performed this test 3 times per leg and averaged values were reported. These measures were used to determine the stronger leg which was tested for all subsequent collection periods to reduce testing time during the race.

## Muscle ultrasound

Ultrasound images of the rectus femoris on the participant's stronger leg (as determined by maximal isometric strength test) were obtained at rest and during isometric contraction. Ultrasound images were acquired using 2D B-mode ultrasonography (Whale Sigma P5, Whale Imaging Inc., Waltham, MA, USA) with a 5–12 MHz frequency, 38mm linear array probe. Participants were positioned supine with knee in 0˚ (neutral), and the hip in 0˚ of abduction and rotation. Padding was used under and around the ankles to standardize the position and to ensure that all participants could comfortably relax. Anatomic landmarks (anterior superior iliac spine (ASIS) and tibial tuberosity) were used to determine image acquisition location and were marked on the participant to ensure consistency for subsequent measures. All images were acquired at 60% of the distance from the tibial tuberosity towards the ASIS. A marker was used to outline the probe location at baseline testing, and participants were instructed to re-apply the outline as necessary to ensure consistent placement of the probe during subsequent image acquisitions. Three static images were acquired in both the resting and contracted conditions to measure MT. For contracted measures, the ultrasound probe was positioned on the rectus femoris, and an image was acquired during MVIC. All image acquisition was performed by two experienced study investigators (Intra and interlimb ICC's 0.929 to 0.979 CI 0.803–0.979, unpublished data). Images were exported to a computer and converted from DICOM to jpeg format using custom Matlab code. ImageJ (National Institutes of Health, Bethesda, MD, USA) was used for offline analysis of MT and analyses were performed by a single investigator. MT was measured as the distance from the superficial to the deep aponeuroses of the rectus femoris. Measurements were taken for each image and averaged to obtain MT at both the resting and contracted conditions. The difference between contracted MT and resting MT (Δ MT) was calculated and reported as both absolute Δ (contracted thickness-resting thickness) and % Δ MT ((contracted thickness-resting thickness)/resting thickness).

## Maximal oxygen uptake (VO$_2$max)

VO$_{2max}$ was assessed during the baseline visit using a constant-speed treadmill protocol with a 2% increase in incline every 2 minutes until exhaustion. The treadmill speed was chosen based on each subject's experience, typical running speed, and heart rate such that VO$_{2max}$ was achieved in 6–12 min. Pulmonary ventilation and expired gas concentrations were analyzed in real time using an automated computerized indirect calorimetry system (COSMED, Rome, Italy). VO$_2$ was considered maximum if a plateau was achieved (increase in VO$_2$ of $<$ 150 ml/min with increased work). In the absence of a clear plateau, tests were verified to meet at least two of the following secondary criteria of maximal effort: a respiratory exchange ratio $>$1.10, a rating of perceived exertion $>$18, and a peak heart rate within 10 beats/min of the age-predicted

maximum. Heart rate was measured during the test using chest strap heart rate monitors (Polar Electro Inc, Lake Success, NY).

### Race day testing

The event was a sanctioned 50-km race in Maryland that consisted of five, 10-km loops. The total elevation change for the course was just under 762 m with no net gain or loss. Weather on race day was sunny with temperatures ranging from 4–8˚C. Race day measures included blood draws, knee extensor MVIC, and muscle ultrasound. Participants were instructed to arrive ~10 hrs fasted and bring their race day meal. Upon arrival on the morning of the race, participants consumed their meal, and a blood draw was performed 30 min after the meal was consumed. Caloric intake is required throughout an ultramarathon, so blood measures were performed in a fed state, instead of fasted, for more accurate reporting of changes in response to exercise. A blood draw was repeated at the 10km mark and within 30min of completing the 50km race. Plasma was isolated from whole blood and was stored at -80˚C until downstream analyses.

Knee extensor MVICs were performed as described above prior to the start of the race, after each 10km, and immediately after completing the 50km race using the same protocol as baseline testing. Ultrasound testing, as described in baseline, was completed immediately post-race. Ultrasound measures at 10km were not included as several participants opted to wear long pants for the race. All measurements performed mid-race took 2–5 minutes which is consistent with time spent refueling during an ultramarathon in non-elite runners.

### 24 hr post-race assessments

Twenty-four hours after race completion, participants returned to the lab and were instructed to bring with them the same pre-race meal that they ate for baseline and race day testing. A blood draw, knee extensor MVIC, and muscle ultrasound were repeated in the fed state following the same procedures and timeframe as described above.

### Blood analyses

Blood samples were successfully obtained at all timepoints from a n = 7 participants. Plasma concentrations of IL-6 and TNF-α were measured in duplicate by multiplex ultra-sensitive sandwich immunoassays (Meso Scale Discovery, Gaithersburg, MD) according to the manufacturer's instructions. Briefly, the plasma sample (or manufacturer-provided standards) and a detection antibody solution were added in sequential steps to 96-well plates pre-coated with capture antibodies in spatially distinct spots. A buffer was then applied to provide an appropriate electrochemiluminescent signal and the plate was read on a Meso Scale Discover SECTOR Imager 2400. The average intra-assay coefficients of variation were 4–9% in these assays. All samples were assayed on the same plate to avoid inter-assay variability.

Plasma concentrations of soluble IL-6 receptor (sIL-6R) were measured in duplicate by enzyme-linked immunosorbent assay (ELISA) (ThermoFisher Scientific, Waltham, MA) according to the manufacturer instructions. This assay has a sensitivity of 0.01 ng/ml. The average intra-assay coefficient of variation was 12.7%.

Plasma calprotectin levels were measured using an ELISA kit (Hycult Biotech, Augst, Switzerland). This kit is designed to detect the heterodimer complex using a capture antibody that recognizes an epitope present on the heterodimer complex but that is not present on either of the monomers. This assay has a sensitivity of ~1 ng/ml. The average intra-assay coefficient of variation was 2.6%, and all samples were assayed on the same plate to avoid inter-assay variability.

As secondary outcomes, blood samples obtained pre-race, 10km, post-race, and 24 hrs post-race were also analyzed for creatine kinase (CK), hematocrit, and leukocytes (Quest Diagnostics, Baltimore, MD) to assess disruption to muscle fibers, dehydration, and an inflammatory response, respectively.

## Statistical analyses

Data were analyzed using SPSS version 22 (IBM, Armonk, NY). Repeated measures ANOVA were run with pairwise comparisons when a main effect of time indicated statistical significance. When data were not normally distributed, non-parametric tests were performed. The criterion for statistical significance was $P \leq 0.05$. Effect sizes were calculated for all statistically significant pairwise comparisons. For comparisons between different time points within the same sex and for the total sample, Cohen's *d* was used. Effect sizes were classified as trivial if $< 0.2$, small if $\geq 0.2$, moderate if $\geq 0.6$, or large if $\geq 1.2$. Results are presented as means ± SD.

## Results

### Demographics

Subject characteristics are presented in Table 1. Eleven individuals (8 men and 3 women) participated in the study, with female participation in this study consistent with overall trends in North America [23] (Table 1). All participants successfully completed the race. All but one participant had previously completed at least 1 ultra-distance event, but overall ultramarathon experience varied widely among participants (ranging from 0–110 previously completed ultra-distance races). Participants had $VO_{2max}$ values consistent with aerobically trained individuals (range 43.5–62.1 ml/kg/min) but were not considered elite [24–27]. Performance in the current race and running/race history can be found in Table 2.

### Rectus femoris muscle thickness

There was no main effect of time for resting MT ($P = 0.345$; Fig 1A). The main effect of time for contracted MT was not significant ($P = 0.09$) although contracted MT was numerically greater 24-hrs post-race compared with baseline ($P = 0.06$, $d = 0.32$; Fig 1B). When assessing absolute Δ MT, there was a significant main effect for time ($P = 0.024$). The absolute Δ MT did not change significantly from baseline to post-race ($P = 0.217$, $d = 0.5$). Compared with baseline values, absolute Δ MT increased by 95% by 24 hrs post-race ($P = 0.016$, $d = 1.1$; Fig 1C). Similarly, when presented as % Δ MT, there was a significant increase at 24 hrs post-race compared to baseline ($11 \pm 11\%$ vs. $22 \pm 8\%$; $P = 0.01$, $d = 1.2$; Fig 1D).

**Table 1. Subject characteristics.**

| | |
|---|---|
| Age (y) | 40 ± 7 |
| BMI (kg/m$^2$) | 24 ± 3.9 |
| $VO_{2max}$ (ml/kg/min) | 51.4 ± 5 |
| Body Fat (%) | 18.9 ± 6.6 |
| Resting SBP (mmHg) | 132 ± 10 |
| Resting DBP (mmHg) | 79 ± 10 |
| Resting MAP (mmHg) | 94 ± 10 |
| Resting HR (bpm) | 62 ± 10 |

BMI, body mass index; $VO_{2max}$, maximal oxygen consumption; SBP, systolic blood pressure, DBP, diastolic blood pressure; MAP, mean arterial pressure; HR, heart rate. Data are means ± SD.

**Table 2. Race performance and history.**

| | |
|---|---|
| Race time (hh:mm:ss) | 6:45.00 ± 0:44.4 |
| Years Exercising Continuously | 16 ± 12 |
| Number of ultras completed total | 22 ± 34 |
| Number of ultras in the past year | 4.0 ± 5.0 |
| Longest ultra-distance completed (km) | 105.8 ± 55.3 |
| Months since the last ultra-race | 6.1 ± 21.2 |
| Longest run in the last 3 months (km) | 59.2 ± 40.2 |
| Longest run in the last 3 months (min) | 570 ± 1784 |
| Self-reported days of running/week | 5 ± 6 |
| Self-reported average km of running/week | 61.4 ± 30.8 |

Data are means ± SD.

## Knee extensor torque

There was a significant main effect of time for knee extension muscle torque (P = 0.038; Fig 2). There was no significant change in knee extensor quadriceps torque from pre-race to 10km (P = 0.64). Compared with 10km, knee extensor torque declined significantly by 20km, but remained unchanged from 20km through post-race (between -10 and -8% from 20km through post-race; P>0.05 for each timepoint; all timepoints shown S1 Fig). Compared with pre-race values, knee extension torque was significantly reduced post-race (-10%; P = 0.047, $d$ = -0.4). At 24hrs post-race, knee extensor torque remained similar to post-race values (P = 0.613).

## Blood analyses

The main effect for time (P<0.001) was significant for hematocrit (Fig 3A). Compared with pre-race, hematocrit was not significantly different post-race (P = 0.062, $d$ = -0.6) but was significantly lower 24hrs post-race (P = 0.001, $d$ = -1.5). There was a significant main effect of time (P = 0.003) for leukocyte number (Fig 3B). Compared with pre-race, the first 10km did not elicit significant changes (P = 0.159) but leukocytes were significantly higher post-race (145%; P = 0.018, $d$ = 2.9). There was a 57% decline in leukocyte number between post-race and 24hrs post-race (P = 0.018, $d$ = -3.0) and no differences between pre-race and 24hr post-race numbers (P = 0.138).

There was a significant main effect of time (P = 0.001) for CK (Fig 4A). CK concentrations rose steadily from pre-race to 10km (40%; P = 0.012, $d$ = 0.6), post-race (323%, P = 0.012, $d$ = 1.9), and 24hrs post-race (718%; P = 0.005, $d$ = 1.0). There was an 98% increase in CK from post-race to 24hrs post-race (P = 0.05, $d$ = 0.6). There was a significant main effect of time (P<0.001) for plasma IL-6 (Fig 4B). Compared with pre-race, IL-6 concentrations increased by 302% after 10km (P = 0.012, $d$ = 1.4) and by 2598% by post-race (P = 0.018, $d$ = 3.9). At 24hrs post-race, IL-6 levels declined substantially from post-race levels but remained slightly elevated compared with pre-race (P = 0.047, $d$ = 1.0). There was no significant change in sIL-6R (P = 0.241; Fig 4C).

The main effect of time (P = 0.001) was significant for calprotectin concentration (Fig 4D). Compared with pre-race, calprotectin levels increased by 50% by 10km (P = 0.017, $d$ = 1.7) and 68% at the post-race timepoint (P = 0.018, $d$ = 2.4). By the 24h post-race timepoint plasma calprotectin concentrations were significantly lower than both 10km (P = 0.012, $d$ = -2.1) and post-race (P = 0.018, $d$ = -3.1) such that levels were no different than those observed at pre-race (P = 0.594). TNF-α concentrations were unaffected by the 50km ultramarathon (main effect P = 0.478; Fig 4E).

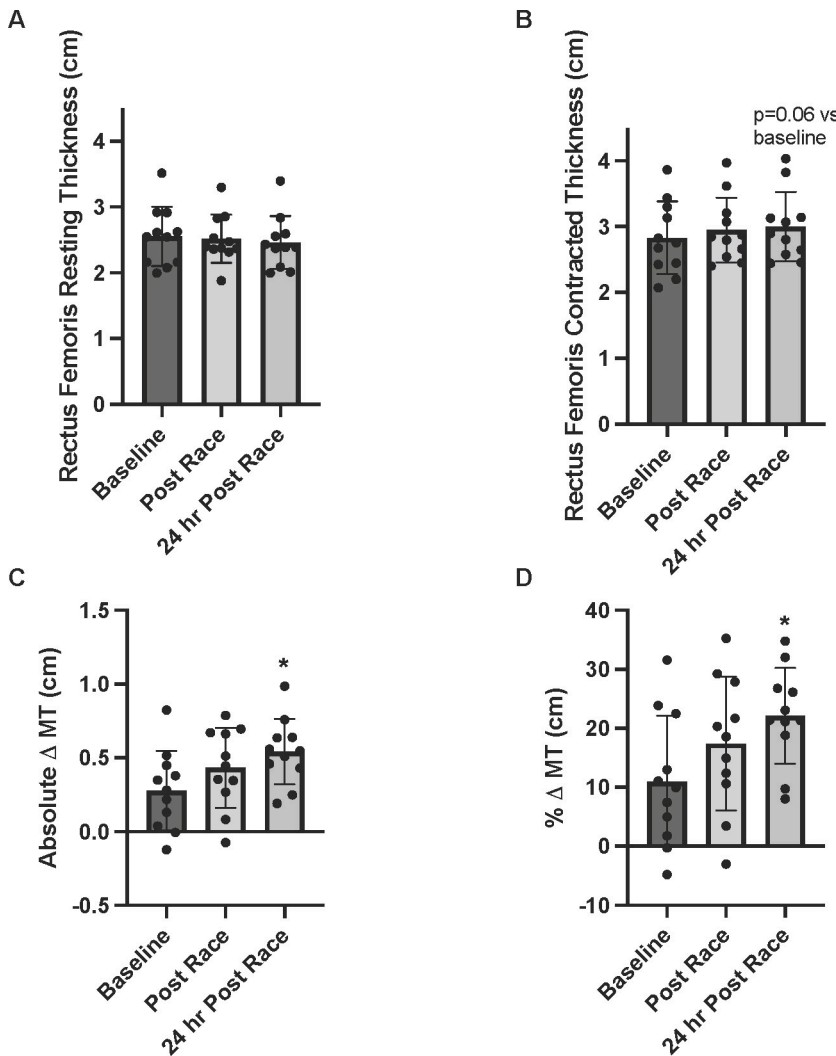

**Fig 1. Muscle thickness in response to a 50km race.** Muscle thickness at baseline, post-race and 24 hrs post-race. Muscle thickness was assessed A) in the resting state, B) during isometric contraction (contracted), and the difference between the resting and contracted is depicted as C) the absolute difference in muscle thickness from contracted to resting and D) % difference in muscle thickness from contracted to resting. Three images were acquired for each condition and an average is presented. *indicates statistically significant than baseline (P ≤ 0.05). Data are reported in means ± SD with data points representing individual participants.

## Discussion

In this study, we present the first report of running-related increases in Δ MT from resting to contracted conditions, despite no changes in resting MT, at 24-hrs post-50km. We also report declines in knee extensor torque and elevations in CK concentrations that appear to remain at the 24-hr post-race timepoint, and acute increases in leukocyte number, plasma IL-6, and calprotectin concentrations that return to pre-race levels within 24 hrs.

While prolonged low-load exercise protocols contribute to muscular swelling or edema [28], this was not observed as greater resting rectus femoris MT following the 50km in our study. Elevation changes, especially elevation loss, can increase muscle swelling and inflammation due to the repeat eccentric contractions that occur during a race with large elevation loss [29]. Common to ultramarathon running, many races exceed 3,000 m [3,11,22,30,31] in

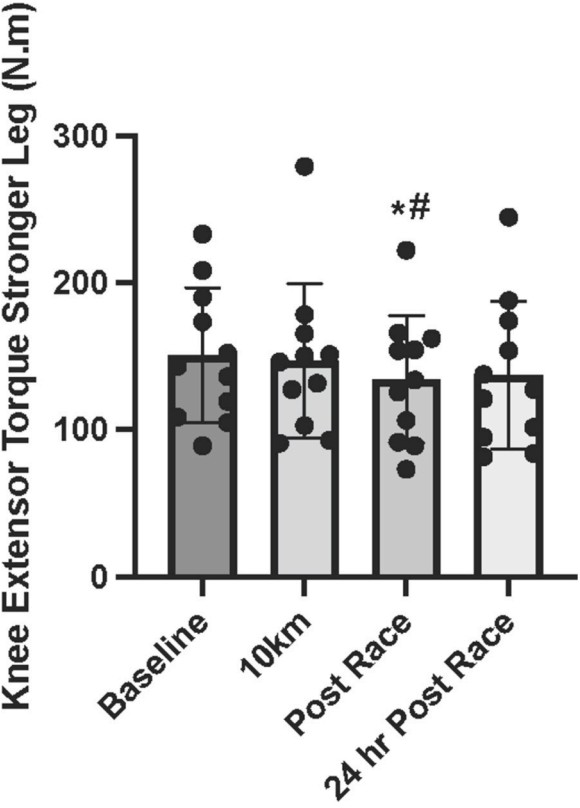

**Fig 2. Knee extensor torque in response to a 50km race.** Knee extensor muscle torque at pre-race, 10km, post-race, and 24 hr post-race. *indicates statistically significant from pre-race, # indicates statistically significant from 10km (P ≤ 0.05). Data are reported in means ± SD with data points representing individual participants.

elevation change over the course of the race. However, by comparison, the current study's race had a total elevation change just under 762 m with no net gain or loss. We did observe progressive increases in CK concentration during and up to 24hrs post-race. The absolute concentrations reported in the current study place the 24hr post-race concentrations just above the clinical standard for active individuals indicating some degree of insult to the muscle fibers [32]. However, studies reporting results from a 67 km race with greater change in elevation (4,500m) have reported 1289% and 1447% increases in CK post-race and 24hrs post-race, respectively (vs. 328% and 718%, respectively, in the current study) with absolute concentrations twice as high as clinical reference values for muscle damage [33].

When compared with baseline, we observed a significantly higher absolute and % Δ MT at 24hrs post-race. To our knowledge, this is the first account of MT being reported as the Δ MT from resting to contracted conditions in this non-elite ultrarunning population. The 24-hr post-race increase in Δ MT is consistent with other reports of delayed onset of muscle soreness [29,34] and is supported by the clinically meaningful increase in CK at 24hrs post-race. Importantly, muscle damage and associated peaks in CK are most apparent several days after the initiating event. Thus, we were not able to distinguish whether the findings in our study are due to muscle damage or other fatigue-related factors. Increases in several of the inflammatory markers observed immediately post-race may play a role in the tightly regulated muscle inflammatory response to this type of exercise. In support of this potential temporal response, Peake et al. found that IL-6 peaked within 1 hr after downhill running, returning to resting levels by 24-hrs post exercise [34]. In comparison, they found that CK displayed a delayed

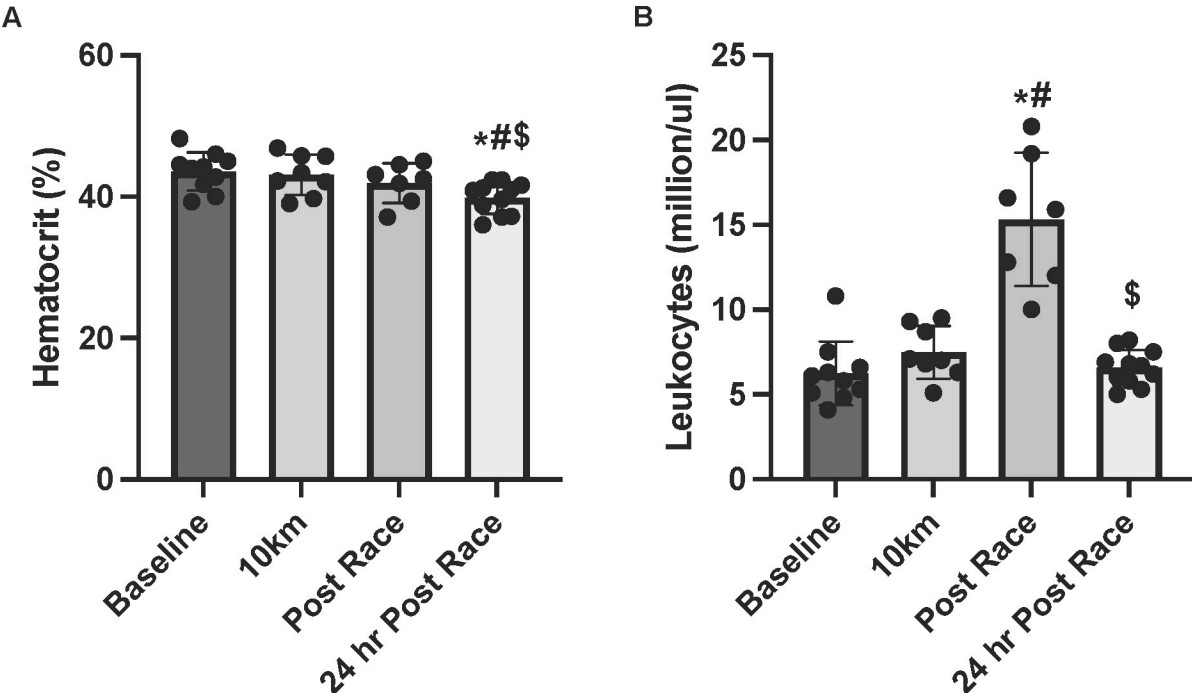

**Fig 3. Blood hematocrit and leukocyte levels.** Blood A) hematocrit level, and B) leukocyte number at pre-race, 10km, post-race, and 24 hr post-race. *indicates statistically significant from pre-race, # indicates statistically significant from 10km, $ indicates statistically significant from post-race (P ≤ 0.05). Data are reported in means ± SD with data points representing individual participants.

response, peaking at 24-hrs post exercise [34]. In the present study, we observed a similar response pattern with leukocyte number, IL-6, and calprotectin increasing significantly post-race but resembling resting values at 24hrs post-race, while CK continued to increase up to 24hrs post-race. The inflammatory response to exercise is highly variable and appears dependent on the extent of the muscle impairment [29]. In our observation, inflammation exhibited post-race may contribute to altered skeletal muscle recruitment patterns at 24-hrs post-race such that over-recruitment [35], as evidenced by greater contracted MT, is required to produce the same force when asked to maximally contract the quadricep [16,35]. This would exhibit as greater Δ MT 24-hrs post-race. Future studies including EMG while assessing MT in the contracted state are needed to confirm this hypothesis. Alternatively, while our results suggest a potential temporal role of post-race inflammation on elevations in MT 24-hrs post-race, we cannot exclude other explanations. For instance, other studies have documented associations between elevated intramuscular fluid pressure and delayed onset of muscle soreness following repeated eccentric activity [19]. This may be more evident in the muscle during the contracted state, causing the increases in ΔMT observed 24-hrs post-race. While the exact mechanisms contributing to changes in ΔMT are not well understood, our findings substantiate the inclusion of MT in both the resting and contracted state to allow for greater insight into changes in muscle morphology in response to acute exercise.

Inflammatory factors play a pivotal role in muscle repair and remodeling in response to exercise [29]. In the present study we found that plasma IL-6 levels increased substantially throughout the 50km while sIL-6 receptor levels were not significantly affected. Although still slightly elevated, IL-6 levels were nearly back to baseline by 24hrs post-race. IL-6 is consistently elevated in response to exercise even in the absence of muscle damage [36], while elevations in sIL-6R are variable [7]. No changes in sIL-6R suggest that IL-6 is acting primarily through its

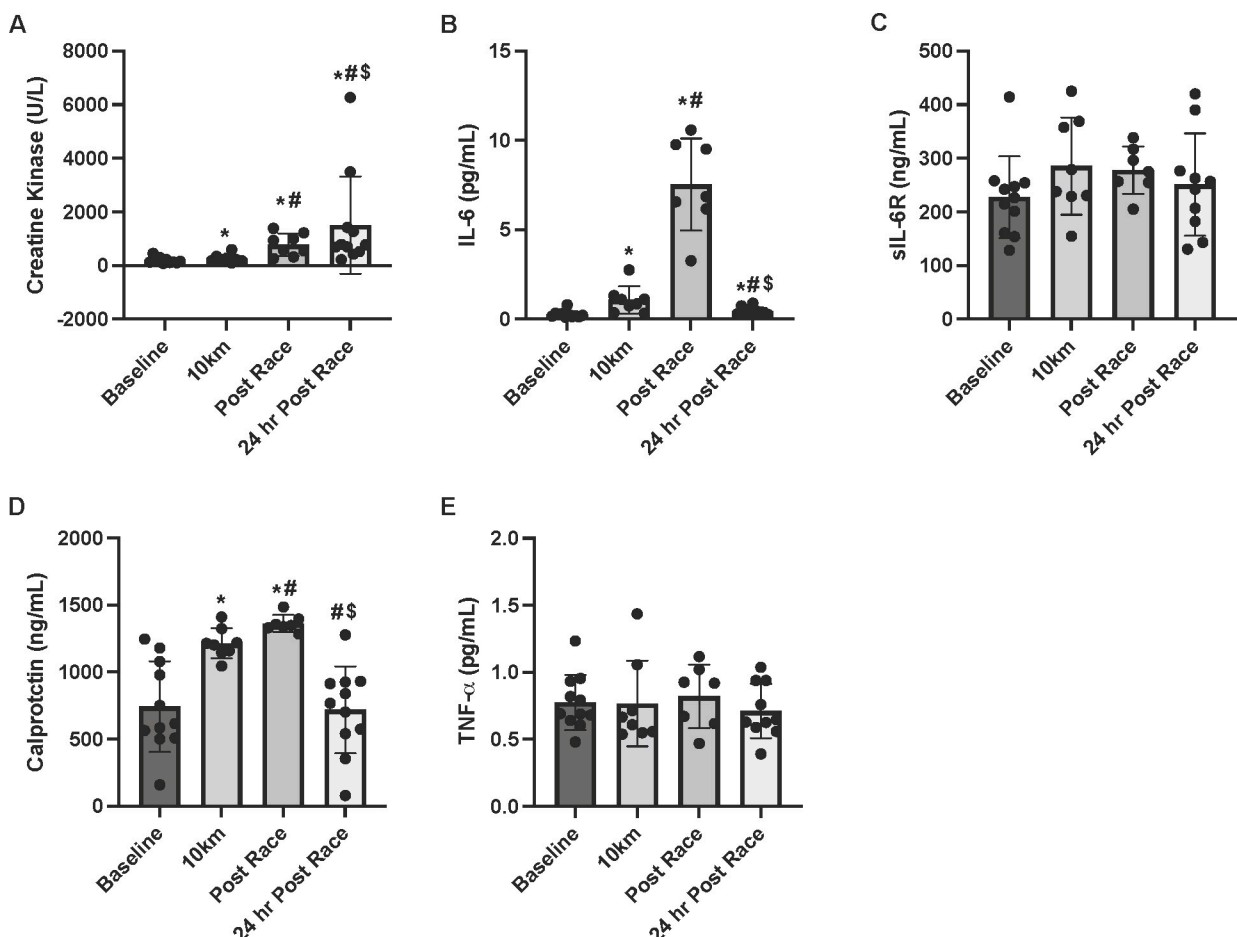

**Fig 4. Plasma markers of inflammation.** Plasma concentrations of A) Creatine kinase B) IL-6, C) sIL-6R, D) calprotectin, and E) TNF-$\alpha$ at pre-race, 10km, post-race, and 24 hr post-race. *indicates statistically significant from pre-race, # indicates statistically significant from 10km, $ indicates statistically significant from post-race (P $\leq$ 0.05). Data are reported in means ± SD with data points representing individual participants.

membrane-bound receptor [8]. Through this pathway, IL-6 most likely aids in fat oxidation and glucose disposal to promote recovery post-exercise [36]. IL-6 and muscle contractions also promote the release of calprotectin from skeletal muscle [37] which was observed in the present study during and following the 50km race. The specific role of calprotectin in response to exercise is not completely understood, but it has been found to activate cellular pathways promoting inflammation [13,37]. Inflammation induced by elevated concentrations of IL-6 and calprotectin post-race may have a collective effect on motor recruitment patterns [7] or contraction-induced swelling [37] that exhibit as physiological phenotype (i.e. greater absolute and % $\Delta$ MT) 24 hrs post-race. In contrast to previous studies, we did not detect any differences in plasma TNF-$\alpha$ levels in response to the 50km race. This may be explained by the relatively short distance of this ultramarathon event in comparison to others [3,11], with TNF-$\alpha$ requiring a greater physiological stimulus than the current race provided.

Inflammation is associated with reduced function and muscle strength [15,29,38]. Accordingly, we observed a decline in muscle strength by the 20km mark that was not evident after the completion of 10km. This strength impairment remained until post-race and, although partially recovered from a numerical standpoint, 24-hr post-race values were not significantly different than immediately post-race. This decline in muscle strength is consistent with the

findings of others post-ultramarathon [15]. Some studies have found that reductions in muscle force production may be present for a up to 2 weeks following longer or more mountainous terrain races [39,40]. Both central and peripheral fatigue can explain decrements in force production following an ultramarathon [16,40] and have been shown to persist for well over 30 minutes post-exercise in endurance events lasting several hours [35]. While not directly assessed in this study, these past findings suggest that it is likely that both central (motor unit recruitment) and peripheral fatigue (metabolite buildup and/or impaired excitation-contraction coupling) contribute to reduced post-race knee extensor torque and may also be influenced by inflammation. Specifically, inflammatory factors including IL-6 [7] can inhibit neural drive by interfering with afferent feedback from exercising muscles [16,41]. In our study, this may be exhibiting as 24 hr post-race declines in muscle strength and compensatory increases in ΔMT representing altered motor unit recruitment [7,16,41]. It is worth noting that the muscular strength tests performed in the present study were isometric. While this assessment is clinically applicable, assessments of muscle strength that include muscle shortening and lengthening more like the act of running may provide a more appropriate challenge to detect persistent changes in muscle strength.

We found that hematocrit levels were slightly reduced 24hrs post-race, likely due to rehydration practices. Although it is possible that this may have influenced our findings, no changes in resting MT, along with elevations in CK indicate that the differences in Δ MT noted 24hrs post-race are indicative of muscle changes, and not solely rehydration. The rectus femoris has been reported to have the least amount of change of the quadriceps muscles during races with heavy eccentric loads, as it sustains the least stimulation during downhill running [42]. Thus, while the rectus femoris is appropriate to assess in this race due to minimal elevation change, other quadriceps muscles should be included for studies in which races are performed with great elevation loss. Findings from this study provide insight into the muscle inflammatory response to ultramarathon running in recreational runners. Differences in race duration, intensity, and elevation change do not allow for direct comparisons between our findings and those from elite runners. Future studies should include both recreational and elite runners to determine whether findings are consistent in individuals with greater fitness levels and training.

While our data support the use of ultrasound as a field-based measure of muscle morphology, this study did come with some limitations. We were limited in our ability to assess MT mid-race due to the race occurring in cold temperatures and most participants choosing to run in long pants. Also, the rectus femoris MT was only assessed in one location. Image acquisition of different areas of muscle and/or other muscles would provide a more comprehensive assessment of muscle responses to ultramarathon running. Furthermore, although participants were coached to perform a maximal isometric contraction for the "contracted" measure, we were not able to confirm a specific level of muscle activation nor that contractions were maximal due to the time constraints and logistics of mid-race testing. As runners often travel long distances to race destinations our participants were not available for testing beyond the 24hr post-race timeframe. Future studies should include assessments beyond 24 hrs post-race to determine the role of muscle damage and the recovery timeframe necessary for skeletal muscle remodeling and resolution of inflammation. Finally, while there is a documented relationship between inflammation and muscle function [15,16,29], we cannot mechanistically conclude that the inflammatory factors we assessed were involved in reduced knee extensor torque or alterations in Δ MT. Correlations between the measured inflammatory factors and functional outcomes were not possible due to limited blood samples, but should be included in future studies to better explain these relationships. Additionally, future studies should expand the inflammatory profile examined and include potential anti-inflammatory cytokines for a more comprehensive understanding of the inflammatory process and associated mechanisms.

Despite these limitations, we determined that portability of ultrasound allows for assessments of MT in longer races and throughout ultra-distance events to detect changes to the muscle in real time. In this study, we document changes in MT in response to prolonged, submaximal muscle contractions. Future studies will investigate MT in response to other stimuli with varying degrees of duration and intensity with the long-term goal of informing mitigation strategies to minimize musculoskeletal damage that may impair performance. Furthermore, ultrasound assessments are quick, cost-effective, and may be used at aid stations without any substantial delays in race time. We were able to complete all ultrasound assessments in less than 5 minutes. This argues for inclusion of ultrasound assessments in future studies to assess muscle morphology in field setting where traditional imaging modalities may be impractical.

In the present study, all participants completed the race, and no participants reported any injuries, suggesting that any muscle impairment suffered was part of a typical remodeling response. There was a range of experience for the runners in this study. Most had approximately 16 years of experience running, the average time spent training was similar across all individuals, and while three of the finishers completed the race in the top 50%, none were of elite status based on times and placement in this or previous races. Thus, the participants in this study were representative of the average non-elite individual who partakes in these events. Furthermore, the range of finish times in this study (5:40.2–8:04.8) is more representative of the average field of ultramarathoners than previously published literature [22,31,39,42]. Although the 50 km distance is shorter than most previously reported work on ultramarathons [3,11,22], this race distance reflects that of the greatest participation rates among all ultramarathon events [2]. Thus, while it may be difficult to make direct comparisons between this and other studies, a strength of this study is that it builds upon the limited body of literature investigating the effects of participation in the most popular ultramarathon distance.

In summary, this study reports delayed increases in absolute and % Δ MT from resting to contracted conditions at 24 hrs post-race, which are preceded by reductions in knee extensor torque and elevations in leukocyte number, plasma IL-6, and calprotectin. IL-6 and calprotectin were released in a similar pattern in response to the 50km. Collectively, these findings indicate the presence of some residual strength impairments and morphological changes to the rectus femoris 24-hrs following a 50km ultramarathon in recreational runners. Results from this study have relevant clinical applicability for understanding the muscle inflammatory response in a population and race distance that is reflective of most ultramarathon participants. It not uncommon for recreational ultramarathon runners to participate in stage races, back-to-back long runs, or weekly marathon and 50k distances [2]. Thus, these findings may be used by athletes, coaches, and trainers to optimize recovery strategies for skeletal muscle health, function, and inflammation when partaking in subsequent prolonged running events.

## Supporting information

**S1 Fig. Knee extensor torque at each 10km timepoint over 50km.** Knee extensor muscle torque at baseline, 10k, 20k, 30k, 40k, post-race, and 24 hr post-race. *indicates statistically significant from baseline, # indicates statistically significant from 10k ($P \leq 0.05$). Data are reported in means ± SD.
(TIF)

## Acknowledgments

We thank the participants for volunteering their time for this study. We thank the race director for allowing our team to collect data at their event.

**Declarations**

**Consent to participate and publish.** Informed consent was obtained from all individual participants included in the study. All individuals provided consent to publish results from this study.

## Author Contributions

**Conceptualization:** Rian Q. Landers-Ramos, Sushant M. Ranadive, Steven J. Prior, Odessa Addison.

**Data curation:** Rian Q. Landers-Ramos, Kathleen Dondero, Christa Nelson, Sushant M. Ranadive, Steven J. Prior, Odessa Addison.

**Formal analysis:** Rian Q. Landers-Ramos, Odessa Addison.

**Investigation:** Rian Q. Landers-Ramos, Kathleen Dondero, Christa Nelson, Sushant M. Ranadive, Steven J. Prior, Odessa Addison.

**Methodology:** Rian Q. Landers-Ramos, Kathleen Dondero, Christa Nelson, Sushant M. Ranadive, Steven J. Prior, Odessa Addison.

**Project administration:** Rian Q. Landers-Ramos, Kathleen Dondero, Odessa Addison.

**Resources:** Rian Q. Landers-Ramos, Sushant M. Ranadive, Steven J. Prior, Odessa Addison.

**Supervision:** Rian Q. Landers-Ramos, Steven J. Prior, Odessa Addison.

**Validation:** Christa Nelson, Odessa Addison.

**Writing – original draft:** Rian Q. Landers-Ramos, Odessa Addison.

**Writing – review & editing:** Rian Q. Landers-Ramos, Kathleen Dondero, Christa Nelson, Sushant M. Ranadive, Steven J. Prior, Odessa Addison.

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
