## [Decision Letter · Decision Letter 0]

1 May 2022

PONE-D-22-07751Muscle thickness and inflammation during a 50km ultramarathon in recreational runnersPLOS ONE

Dear Dr. Landers-Ramos,

Thank you for submitting your manuscript to PLOS ONE. After careful consideration, we feel that it has merit but does not fully meet PLOS ONE’s publication criteria as it currently stands. Therefore, we invite you to submit a revised version of the manuscript that addresses the points raised during the review process.

Both reviewers found merit in the work but have suggested changes for improvement. The authors also need to provide the data used in the study as a supplementary table. This is a PLOS ONE requirement for publication.

We look forward to receiving your revised manuscript.

Kind regards,

Jeremy P Loenneke

Academic Editor

PLOS ONE

Journal Requirements:

Reviewers' comments:

Reviewer's Responses to Questions

**Comments to the Author**

1. Is the manuscript technically sound, and do the data support the conclusions?

Reviewer #1: No

Reviewer #2: Partly

2. Has the statistical analysis been performed appropriately and rigorously? 

Reviewer #1: Yes

Reviewer #2: Yes

3. Have the authors made all data underlying the findings in their manuscript fully available?

Reviewer #1: No

Reviewer #2: Yes

4. Is the manuscript presented in an intelligible fashion and written in standard English?

Reviewer #1: Yes

Reviewer #2: Yes

5. Review Comments to the Author

Reviewer #1: Summary: The authors investigated the alterations in muscle thickness (MT), muscle strength, and blood levels of inflammatory cytokines (IL-6,sIL-6R, TNF-�, and calprotectin) following a 50km race in non-elite athletes. The main findings include significantly increased ΔMT at 24-hrs post-race, reduction in knee extensor strength, elevated CK post-race and 24-hr post-race. In addition, the blood leukocyte number, IL-6, and calprotectin increased acutely post-race and returned to baseline levels at 24-hr psot-race.

Major comments:

The biggest problem of this manuscript is that all the tables are missing. This reviewer cannot find any tables in the submitted file.

One of the major limitations of this manuscript is the weak introduction section, which makes the novelty and relevance of the study weak. For example, what are the functions of IL-6, sIL-6R, calprotectin? Why are they important to measure in this study? What is the difference between IL-6 and sIL-6R, and why are they both needed to measure instead of either one? What is the rationale of measuring muscle thickness? What does the ΔMT mean vs. MT physiologically? What is the rationale of analyzing ΔMT vs. just MT data? How is MT/ ΔMT related to muscle swelling or inflammation?

Detailed comments:

1. Lines 259,261, 266: Where are the Table 1 & Table 2?

2. Line 265: “ were not considered elite.”

- What is the definition of eliteness?

3. Line 273: “non-significant increase of 55% from baseline to post-race (P= 0.217, d = 0.5)” .

- If there are no significant changes, it should not be described as non-significant changes.

4. Line 353: “In our observation, post-race inflammation may contribute to delayed alterations in muscle recruitment exhibited as greater ΔMT noted 24-hrs post-race”

- How is ΔMT related to muscle recruitment? This should be explained in the introduction section. It is confusing that greater ΔMT indicates muscle recruitment alterations.

5. Line 360-361: “the rectus femoris has been reported to have the least amount of change of the quadriceps muscles during races”

- Then why chose it as the muscle group for the measurements?

6. Lines 401-402: “This study was designed to determine the practicality of using ultrasound as a field-based measure..”

- The results don’t support this. This reviewer couldn’t tell that the data showed the practicality of using ultrasound as a measure of the inflammation during the race. Needs more explanation/justification.

Reviewer #2: This manuscript describes an experiment examining markers of inflammation during and following a 50Km ultramarathon. The manuscript is well-written and the methods and experimental design are, mostly, appropriate to answer the research questions. Please see my detailed comments below to help strengthen and clarify the manuscript.

Major Comments

1. The authors do not comment on the development of peripheral or central fatigue and how that might have impacted inflammation, strength, and performance. The authors, rightly in my mind, discuss the idea of muscle damage that may have occurred during the race and how this could have impacted MT and strength. However, the loss of force production during the race may simply have been due to fatigue. Is there a relationship between fatigue and inflammation? If so, how might this have impacted the results?

2. I appreciate the effort of the authors to conduct field-based research given its difficulties. I am curious as to why participants were only retested at 24 hours post? Muscle damage often manifests for days to week after eccentric exercise with inflammation due to muscle damage often peaking 7-10 days after the insult. Given that force production had essentially returned to baseline levels and that no measure of muscle soreness was collected it is very difficult to conclude that appreciable damage actually occurred--indeed the fact that force was recovered 24hr post strongly suggests that decrements during exercise were simply due to fatigue. Thus any change in blood markers of inflammation or MT may or may not be due to damage.

3. Do the authors feel that changes in inflammatory markers during exercise could simply have been the "normal" respsone and may actually not play any role in changes in MT, force, or performance?

4. I know the sample size is small, but reporting correlations between changes in MT, force, and blood markers could be help for beginning to understand the relationship among them.

5. Are there alternative explanations for the change in MT during exercise beyond inflammation? Changes in intramuscular pressure as well as changes in oncotic pressure due to metabolism could easily function to drive water out of blood and the interstitial space and into skeletal muscle fibers. I feel this should be discussed. The fact that it remains elevated 24 hours later is suggestive of a longer lasting effect, which could be from inflammation.

6. I think the idea that understanding how acute changes in MT and inflammation, which could be tracked by ultrasound, could lead to increased risk of musculoskeletal injury needs to be further developed. Is there any evidence to support such a suggestion?

Specific Comments

1. p13, ln286 - please do not refer to non-significant results as a "tendency"

2. p16, ln 353 - how can post race inflammation be related to changes in recruitment as indicated by MT measures? This seems very tenuous at best.

3. p19, ln 424 - I would caution the authors about suggesting muscle damage is part of a typical remodeling response. Damage in a mechanical event, likely due to eccentric exercise. Following the damage, there will likely be repair and perhaps remodeling that will occur.

6. PLOS authors have the option to publish the peer review history of their article (what does this mean?). If published, this will include your full peer review and any attached files.

Reviewer #1: No

Reviewer #2: No

---

## [Author Response · Author response to Decision Letter 0]

14 May 2022

Editor comments:

Formatting has been edited as per the requirements listed above.

Tables have been added to the main manuscript.

If there are ethical or legal restrictions on sharing a de-identified data set, please explain them in detail (e.g., data contain potentially sensitive information, data are owned by a third-party organization, etc.) and who has imposed them (e.g., an ethics committee). Please also provide contact information for a data access committee, ethics committee, or other institutional body to which data requests may be sent.

This has been addressed in our new cover letter.

The ethics statement has been moved to the methods section.

This has been corrected.

Reviewer comments (please also see attached response to reviewer's):

Reviewer #1: 

1.The biggest problem of this manuscript is that all the tables are missing. This reviewer cannot find any tables in the submitted file.

A per journal requirements, tables have been re-located directly after the paragraph in which they are first cited.

2. One of the major limitations of this manuscript is the weak introduction section, which makes the novelty and relevance of the study weak. For example, what are the functions of IL-6, sIL-6R, calprotectin? Why are they important to measure in this study? What is the difference between IL-6 and sIL-6R, and why are they both needed to measure instead of either one? What is the rationale of measuring muscle thickness? What does the ΔMT mean vs. MT physiologically? What is the rationale of analyzing ΔMT vs. just MT data? How is MT/ ΔMT related to muscle swelling or inflammation?

We have significantly revised the introduction to improve clarity. In keeping with journal guidelines regarding the length of the introduction, further details about the respective roles of each inflammatory factor measured and how we interpret these factors affecting the other results in this study have been included in the discussion.

We have also clarified the novelty and importance of including both the contracted MT and ∆ MT measures as a more comprehensive picture of changes in muscle morphology beyond just resting measures which have been the major focus of previous studies.

Line 76

“Ultramarathon running, specifically, has been found to elicit large increases in IL-6 (3–5) which has numerous physiological roles. When bound to its soluble receptor (sIL-6R), IL-6 acts through a trans-signaling pathway to promote cellular apoptosis (6). Other inflammatory factors such as tumor necrosis factor (TNF)-� (3, 7–9), and calprotectin (10, 11) are elevated after prolonged running and are involved in the activation and breakdown of damaged cells (10, 12).”

Line 85

“Further, inclusion of MT measures during muscle contraction may provide more insight into changes in muscle morphology post-exercise as it may reflect shifts in intramuscular fluid pressure (17), or altered muscle recruitment, especially when used in conjunction with measures of muscle strength (18). Therefore, reporting MT under both resting and contracted conditions as well as ∆ MT to demonstrate changes relative to resting conditions provides a more comprehensive picture of acute changes in muscle morphology in response to ultramarathon running.”

Detailed comments:

1. Lines 259,261, 266: Where are the Table 1 & Table 2? 

A per journal requirements, tables have been re-located directly after the paragraph in which they are first cited.

2. Line 265: “ were not considered elite.”

- What is the definition of eliteness?

We define elite status as determined by measured VO2max values in comparison to several well-established reports of VO2max in competitive endurance athletes across various ages (Heath et al. JAP 1981; Robinson et al., JAP 1976; Foster et al. Eur J Appl Physiol and Occup Physiol 1978; Bergh et al. Med Sci Sports 1978). Our average VO2max value was ~50 ml/kg/min and values of 70 ml/kg/min or higher are frequently reported in elite endurance athletes. We have added these references to the manuscript on line 276.

3. Line 273: “non-significant increase of 55% from baseline to post-race (P= 0.217, d = 0.5)” .

- If there are no significant changes, it should not be described as non-significant changes.

This has been corrected.

4. Line 353: “In our observation, post-race inflammation may contribute to delayed alterations in muscle recruitment exhibited as greater ΔMT noted 24-hrs post-race”

- How is ΔMT related to muscle recruitment? This should be explained in the introduction section. It is confusing that greater ΔMT indicates muscle recruitment alterations.

Clarification on contracted and ΔMT have been added to the introduction as indicated above.

Inflammation immediately post-race (supported by elevations in several inflammatory cytokines) may influence neuromuscular recruitment (Millet et al. 2018 APNM) and alter motor unit responsiveness leading to less efficient motor unit contraction (greater recruitment to perform the same task). Indeed, we found that contracted MT exhibited a delayed elevation at 24-hrs post-race that, while not statistically significant, was substantially greater than immediately post-race and in the absence of resting MT changes, appears to be a potential contributor to our observed changes in ΔMT. The influence of inflammation on muscle recruitment has been added to the introduction:

In response to feedback from both reviewers, we have elaborated on this theory in the discussion and also posed an alternative explanation to our observed changes in ΔMT.

Line 392

“In our observation, inflammation exhibited post-race may contribute to altered skeletal muscle recruitment patterns at 24-hrs post-race such that over-recruitment (32), as evidenced by greater contracted MT, is required to produce the same force when asked to maximally contract the quadricep (12, 32). This would exhibit as greater ∆ MT 24-hrs post-race. Future studies including EMG while assessing MT in the contracted state are needed to confirm this hypothesis. Alternatively, while our results suggest a potential temporal effect of inflammation on elevations in MT 24-hrs post-race, we cannot exclude other explanations. For instance, other studies have documented associations between elevated intramuscular fluid pressure and delayed onset of muscle soreness following repeated eccentric activity (15). This may be more evident in the muscle during the contracted state, causing the increases in ΔMT observed 24-hrs post-race. While the exact cause of ΔMT is not well understood, our findings substantiate the inclusion of MT during the contracted state as it provides insight into changes in muscle morphology post-race that were not detected in the resting state. However, as this is one of the first studies to utilize ultrasound in both resting and contracted conditions, further research is needed to fully understand what is causing alterations in MT during contraction.”

5. Line 360-361: “the rectus femoris has been reported to have the least amount of change of the quadriceps muscles during races”

- Then why chose it as the muscle group for the measurements?

Since the rectus femoris is superficial we selected this muscle because we were able to quickly acquire images in the field with a single ultrasound probe across various body sizes in both the resting and contracted state. 

Originally, we stated “the rectus femoris has been reported to have the least amount of change of the quadriceps muscles during races with heavy eccentric loads, as it sustains the least stimulation during downhill running (32) This suggests that for studies with greater elevation changes, other quadriceps muscles should be included in the assessment.” Our original intent was to discuss our findings in a race with little elevation change, in comparison to many of the previous studies that have examined runners after higher levels of elevation change. As noted in paragraph 2 of the discussion, line 366, the race performed for this study had only 762 meters of elevation change over ~31 miles with no net loss, making the eccentric loads sustained in this study minimal. While the rectus femoris is appropriate for our study that had minimal elevation change, if races with greater elevation change, and therefore, greater eccentric loading are to be included, it would be important to include other quadriceps muscles as well. 

To clarify our original statement, we now write:

(Line 413)

 “Thus, while the rectus femoris is appropriate to assess in this race due to minimal elevation change, other quadriceps muscles should be included for studies in which races are performed with great elevation loss.”

6. Lines 401-402: “This study was designed to determine the practicality of using ultrasound as a field-based measure..”

- The results don’t support this. This reviewer couldn’t tell that the data showed the practicality of using ultrasound as a measure of the inflammation during the race. Needs more explanation/justification.

We appreciate the question and have rephrased this sentence to better represent the point that, while ultrasound can be used to assess muscle swelling, as a field-based measure it comes with some limitations.

Line 461

“While our data support the use of ultrasound as a field-based measure of muscle morphology, this study did come with some limitations.”

Reviewer #2: 

1. The authors do not comment on the development of peripheral or central fatigue and how that might have impacted inflammation, strength, and performance. The authors, rightly in my mind, discuss the idea of muscle damage that may have occurred during the race and how this could have impacted MT and strength. However, the loss of force production during the race may simply have been due to fatigue. Is there a relationship between fatigue and inflammation? If so, how might this have impacted the results?

We agree that commenting on peripheral and central fatigue is a valuable addition to this manuscript and thank this reviewer for the opportunity to include this in our discussion. We have added content to the discussion regarding the effects of central and peripheral fatigue as well as the relationship between inflammation and fatigue.

Line 448

“Both central and peripheral fatigue can explain decrements in force production following an ultramarathon (14, 39) and have been shown to persist for well over 30 minutes post-exercise in endurance events lasting several hours (34). While not directly assessed in this study, these past findings suggest that it is likely that both central (motor unit recruitment) and peripheral fatigue (metabolite buildup and/or impaired excitation-contraction coupling) contribute to reduced post-race knee extensor torque and may also be influenced by inflammation. Specifically, inflammation can inhibit neural drive by interfering with afferent feedback from exercising muscles (14, 40).”

2. I appreciate the effort of the authors to conduct field-based research given its difficulties. I am curious as to why participants were only retested at 24 hours post? Muscle damage often manifests for days to week after eccentric exercise with inflammation due to muscle damage often peaking 7-10 days after the insult. Given that force production had essentially returned to baseline levels and that no measure of muscle soreness was collected it is very difficult to conclude that appreciable damage actually occurred--indeed the fact that force was recovered 24hr post strongly suggests that decrements during exercise were simply due to fatigue. Thus any change in blood markers of inflammation or MT may or may not be due to damage.

We thank the reviewer for this comment. Not including measures beyond 24-hrs post-exercise is a limitation in our study as noted in line 469. Specifically, many of our participants were not local and were therefore unable to come in for testing on a weekday morning due to work obligations. We agree with this reviewer that, given the timeframe in which we were able to collect post-race data, we cannot conclude that these changes in MT or inflammation are due specifically to muscle damage. Rather, we have selected the terms inflammation and swelling which we feel more accurately represent the findings in this study. We have revisited the manuscript to remove conclusions that imply that our results reflect muscle damage and have added the following sentence to address this point

line 381:

“Importantly, muscle damage and associated peaks in CK are most apparent several days after the initiating event. Thus, we were not able to distinguish whether the findings in our study are due to muscle damage or other fatigue-related factors.”

3. Do the authors feel that changes in inflammatory markers during exercise could simply have been the "normal" response and may actually not play any role in changes in MT, force, or performance?

We have included this possibility in our limitations section. 

Line 473 

“Finally, while there is a documented relationship between inflammation and muscle function (11, 12, 25), we cannot mechanistically conclude that the inflammatory factors we assessed were involved in reduced knee extensor torque or alterations in ∆ MT.”

4. I know the sample size is small, but reporting correlations between changes in MT, force, and blood markers could be help for beginning to understand the relationship among them.

Due to difficulty obtaining full sets of blood samples from participants, our ability to perform correlations between inflammatory factors, MT and force production was limited. To not mislead our readers, we have opted to exclude these correlations from the study. We agree this would be great future direction and have added it into the discussion.

Line 476

“Correlations between the measured inflammatory factors and functional outcomes were not possible due to limited blood samples, but should be included in future studies to better explain these relationships”

5. Are there alternative explanations for the change in MT during exercise beyond inflammation? Changes in intramuscular pressure as well as changes in oncotic pressure due to metabolism could easily function to drive water out of blood and the interstitial space and into skeletal muscle fibers. I feel this should be discussed. The fact that it remains elevated 24 hours later is suggestive of a longer lasting effect, which could be from inflammation.

We thank this reviewer for the added insight into potential increases in ΔMT observed in our study. We have added the following statement to address this in the discussion section.

Line 397: 

“Alternatively, while our results suggest a potential temporal effect of inflammation on elevations in MT 24-hrs post-race, we cannot exclude other explanations. For instance, other studies have documented associations between elevated intramuscular fluid pressure and delayed onset of muscle soreness following repeated eccentric activity (15). This may be more evident in the muscle during the contracted state, causing the increases in ΔMT observed 24-hrs post-race. While the exact cause of ΔMT is not well understood, our findings substantiate the inclusion of MT during the contracted state as it provides insight into changes in muscle morphology post-race that were not detected in the resting state. However, as this is one of the first studies to utilize ultrasound in both resting and contracted conditions, further research is needed to fully understand what is causing alterations in MT during contraction.”

6. I think the idea that understanding how acute changes in MT and inflammation, which could be tracked by ultrasound, could lead to increased risk of musculoskeletal injury needs to be further developed. Is there any evidence to support such a suggestion?

This is the first of several studies in our lab that use ultrasound imaging to detect differences in muscle morphology in response to a variety of muscle stimulation/exercise protocols. Long-term, we hope that by documenting changes in response to these training and competition-based stimuli may be able to help predict injury risk. However, in line with what this review has noted, the connection is not yet clear. We have edited this section in the discussion to better reflect our plans for the future. 

Line 480

“In this study, we document changes in MT in response to prolonged, submaximal muscle contractions. Future studies will investigate MT in response to other stimuli with varying degrees of duration and intensity with the long-term goal of informing mitigation strategies to minimize musculoskeletal damage that may impair performance.”

Specific Comments

1. p13, ln286 - please do not refer to non-significant results as a "tendency"

This has been removed.

2. p16, ln 353 - how can post race inflammation be related to changes in recruitment as indicated by MT measures? This seems very tenuous at best.

While not believed to be the only explanation, there is some evidence that inflammation can interfere with afferent feedback from exercising muscle and alter motor unit responsiveness (Millett et al. APNM 2018). We have added this as a potential explanation. Additionally, as indicated above, we have proposed another potential explanation for the observed changed in MT.

Line 392

“In our observation, inflammation exhibited post-race may contribute to altered skeletal muscle recruitment patterns at 24-hrs post-race such that over-recruitment (32), as evidenced by greater contracted MT, is required to produce the same force when asked to maximally contract the quadricep (12, 32). This would exhibit as greater ∆ MT 24-hrs post-race. Future studies including EMG while assessing MT in the contracted state are needed to confirm this hypothesis.”

3. p19, ln 424 - I would caution the authors about suggesting muscle damage is part of a typical remodeling response. Damage in a mechanical event, likely due to eccentric exercise. Following the damage, there will likely be repair and perhaps remodeling that will occur.

This statement has been removed.

---

## [Decision Letter · Decision Letter 1]

6 Jul 2022

PONE-D-22-07751R1Muscle thickness and inflammation during a 50km ultramarathon in recreational runnersPLOS ONE

Dear Dr. Landers-Ramos,

Thank you for submitting your manuscript to PLOS ONE. After careful consideration, we feel that it has merit but does not fully meet PLOS ONE’s publication criteria as it currently stands. Therefore, we invite you to submit a revised version of the manuscript that addresses the points raised during the review process.

 Both reviewers felt your manuscript has been improved. However, Reviewer 1 felt additional revisions are necessary in order to improve the clarity of the manuscript. Please pay particular attention to those comments for the next revision.

We look forward to receiving your revised manuscript.

Kind regards,

Jeremy P Loenneke

Academic Editor

PLOS ONE

Journal Requirements:

Reviewers' comments:

Reviewer's Responses to Questions

**Comments to the Author**

1. If the authors have adequately addressed your comments raised in a previous round of review and you feel that this manuscript is now acceptable for publication, you may indicate that here to bypass the “Comments to the Author” section, enter your conflict of interest statement in the “Confidential to Editor” section, and submit your "Accept" recommendation.

Reviewer #1: (No Response)

Reviewer #2: All comments have been addressed

2. Is the manuscript technically sound, and do the data support the conclusions?

Reviewer #1: Yes

Reviewer #2: Yes

3. Has the statistical analysis been performed appropriately and rigorously? 

Reviewer #1: Yes

Reviewer #2: Yes

4. Have the authors made all data underlying the findings in their manuscript fully available?

Reviewer #1: Yes

Reviewer #2: Yes

5. Is the manuscript presented in an intelligible fashion and written in standard English?

Reviewer #1: Yes

Reviewer #2: Yes

6. Review Comments to the Author

Reviewer #1: Overview: This reviewer appreciates the revisions and clarifications. However, the introduction and discussion sections still need further improvement.

The detailed comments are listed below.

1. Introduction

1) It is still not very clear why the authors chose IL-6, sIL-6, TNF-alpha, calprotectin as the inflammatory markers to measure. For example, why didn’t the author measure other commonly used inflammatory markers such as CRP, IL-1B?

2) Line 77: “found to elicit large increases in IL-6 [3–5] which has numerous physiological roles”

It would be better to expand this sentence on the role(s) of IL-6, especially its roles closely related to this study.

Line 78: Same issue with sIL-6R. Are there insoluble IL-6Rs? What does “tran-signaling pathway”

3) Lines 86&87: “into changes in muscle morphology post-exercise as it may reflect shifts in intramuscular fluid pressure [17], or altered muscle recruitment”

Please explain these further. For example, what kinds of shifts in intramuscular fluid pressure? What kinds of muscle recruitment alterations can be indicated by the MT?

4) As one of the main outcome measures, CK was not mentioned at all in the introduction section. What is the rationale of CK measurement in this study?

5) Line 98: “men with a higher VO2max (>60 ml/kg/min) compared to those with a VO2max < 55 ml/kg/min”.

Are these values the threshold values to define the elite runners vs. recreational runners? If so, the authors should define these terms somewhere here or in the section of “Participant selection”.

Lines 275&276. What were the largest and smallest VO2max values of the subjects?

2. Discussion

1) Overall, this reviewer still doesn’t understand how the findings demonstrate that MT measurement “may increase sensitivity to muscle changes” (Line 419). This study found that ΔMT (both absolute and relative values) was significantly increased at 24hr-post race. However, inflammatory cytokines like IL-6 were increased immediately post-race, which seemed more sensitive to the muscle changes?

2) Lines 396-406: There were significant changes in the deltaMT, but no significant changes in the MT. Any discussion on that? Does that mean future studies should use deltaMT instead of MT to analyze the muscle changes?

3) Lines 430&431: “The specific role of calprotectin in response to exercise is not completely understood”

Then why did the authors choose to measure calprotectin? Needs to be explained in the introduction.

4) Lines 422-429: The discussion on the role of IL-6 here is very confusing. The greater ΔMT was observed at 24hr post-race when the “IL-6 levels were nearly returned back to baseline”. How does “the greater absolute and % ΔMT observed at 24 hrs post-race” support “IL-6 may aid in muscle regeneration in response to exercise through the classical IL-6 signaling pathway associated with anti-inflammatory properties [6].”? How does this suggest “a delayed inflammatory response”?

5) Line 424: Why sIL-6R levels were not affected? Any discussion on that?

6) Lines 444-446: “The decline in muscle strength… may contribute to …observed increase in ΔMT…”. So, the changes in ΔMT were due to the muscle strength decline? Very confusing.

7) Line 509: “…delayed onset of muscle inflammation…”. What does this mean? Does it mean the muscle inflammatory response only occurred at a later timepoint like 24hr post-race in this study? Needs to be careful with “delayed onset” here because the acute inflammatory response obviously already initiated quickly post-race or even during the race as indicated by the increased IL-6 levels at 10km and post-race.

8) One of the main rationales of this study was to investigate the “trajectory of the inflammatory process in non-elite athletes..”(Line 103). It would be nice if the authors discuss whether their findings matched with those of studies in elite athletes.

9) It was not clear how the findings from this study can help “develop training plans for longer events and design recovery strategies to optimize skeletal…”(Line 513). Pleaser provide more direct connection between them.

Reviewer #2: (No Response)

7. PLOS authors have the option to publish the peer review history of their article (what does this mean?). If published, this will include your full peer review and any attached files.

Reviewer #1: No

Reviewer #2: No

---

## [Author Response · Author response to Decision Letter 1]

13 Jul 2022

1) It is still not very clear why the authors chose IL-6, sIL-6, TNF-alpha, calprotectin as the inflammatory markers to measure. For example, why didn’t the author measure other commonly used inflammatory markers such as CRP, IL-1B?

We appreciate this reviewer’s interest in the other inflammatory markers that may be affected by ultramarathon running. A comprehensive panel inflammatory factors was not possible for this study so we selected factors with established roles in systemic inflammation that have been previously found to be influenced by prolonged exercise and may affect muscle function. The roles of the selected cytokines are now further defined in the introduction. Additionally, we have included a statement in the limitations to address not including a broader panel of inflammatory factors.

Line 76: Ultramarathon running, specifically, has been found to elicit large increases in IL-6 [3–5]. Among its numerous physiological roles, IL-6 may limit motor activity as a protective mechanism in response to fatigue from prolonged or intense exercise [6,7]. Further, when bound to its soluble receptor (sIL-6R), IL-6 acts to promote cellular apoptosis [8] and the inflammatory properties may be amplified [7]. Other inflammatory factors such as tumor necrosis factor (TNF)-� [3,9–11], and calprotectin [12,13] are elevated after prolonged running and are involved in the activation and breakdown of damaged cells, including skeletal muscle cells, through inflammatory signaling pathways [12,14]. Together, these factors may contribute to declines in muscle function (e.g., acute decreases in strength) [15] seen after an ultramarathon event and play a role in subsequent recovery [16].

Line 487: “…future studies should expand the inflammatory profile examined and include potential anti-inflammatory cytokines for a more comprehensive understanding of the inflammatory process and associated mechanisms.”

2) Line 77: “found to elicit large increases in IL-6 [3–5] which has numerous physiological roles”

It would be better to expand this sentence on the role(s) of IL-6, especially its roles closely related to this study. 

Line 78: Same issue with sIL-6R. Are there insoluble IL-6Rs? What does “tran-signaling pathway”

We have expanded the description of the role of IL-6 that are most relevant to this study. IL-6 has both membrane-bound and soluble receptors. Studies have found that the role of IL-6 is dependent on the receptor and subsequent signaling pathway that is elicited (Scheller et al., Biochimica et Biophysica Acta-Molecular Cell Research 2011). We selected sIL-6R as elevations in both IL-6 and sIL-6 receptor would be indicative of trans-signaling pathway and would suggest inflammation and apoptosis. This has been clarified in the introduction.

Line 76: “Ultramarathon running, specifically, has been found to elicit large increases in IL-6 [3–5]. Among its numerous physiological roles, IL-6 may limit motor activity as a protective mechanism in response to fatigue from prolonged or intense exercise [6,7]. Further, when bound to its soluble receptor (sIL-6R), IL-6 acts to promote cellular apoptosis [8] and the inflammatory properties may be amplified [7].”

3) Lines 86&87: “into changes in muscle morphology post-exercise as it may reflect shifts in intramuscular fluid pressure [17], or altered muscle recruitment”

Please explain these further. For example, what kinds of shifts in intramuscular fluid pressure? What kinds of muscle recruitment alterations can be indicated by the MT?

This has been explained in more detail in the introduction:

Line 89: For instance, contracted MT may emphasize intracellular swelling due to ions released from cytoskeletal breakdown after repetitive muscle contractions and resulting in shifts in osmotic pressure gradients [19]. Further, contracted MT may reflect altered muscle recruitment (failure to fully recruit or over-recruitment in response to a standard task).

4) As one of the main outcome measures, CK was not mentioned at all in the introduction section. What is the rationale of CK measurement in this study?

While involved in the inflammatory process, CK is considered a marker of muscle damage and not inflammation and was therefore not included as a main outcome measure. Rather, CK, along with hematocrit and leukocyte, were included as secondary measures to confirm or exclude the existence of muscle damage (CK), dehydration (hematocrit), and an inflammatory response (leukocytes). These outcomes are all included in the discussion as a means of comprehensively interpreting our results. The methods have been clarified to reflect the purpose of including these outcomes.

Line 259: “As secondary outcomes, blood samples obtained pre-race, 10km, post-race, and 24 hrs post-race were also analyzed for creatine kinase (CK), hematocrit, and leukocytes (Quest Diagnostics, Baltimore, MD) to assess disruption to muscle fibers, dehydration, and an inflammatory response, respectively.” 

5) Line 98: “men with a higher VO2max (>60 ml/kg/min) compared to those with a VO2max < 55 ml/kg/min”.

Are these values the threshold values to define the elite runners vs. recreational runners? If so, the authors should define these terms somewhere here or in the section of “Participant selection”.

The VO2max cutoffs that this reviewer is referencing come from a previously published article by Mooren et al. Int J Sports Med, 2006 and was not established by the current authors. In the Mooren et al. paper the authors do not specify the criteria they used to determine this threshold. We did not identify a specific VO2max value in our inclusion criteria. Rather, we performed VO2max tests to define subject characteristics and confirm that the values recorded are consistent with recreational runners using references from multiple previous studies provides in line 282 of our manuscript.

6) Lines 275&276. What were the largest and smallest VO2max values of the subjects?

The lowest recorded VO2max in our study was 43.5 ml/kg/min while the highest was 62.1 ml/kg/min. This range has been added to line 282. 

7) Overall, this reviewer still doesn’t understand how the findings demonstrate that MT measurement “may increase sensitivity to muscle changes” (Line 419). This study found that ΔMT (both absolute and relative values) was significantly increased at 24hr-post race. However, inflammatory cytokines like IL-6 were increased immediately post-race, which seemed more sensitive to the muscle changes?

We agree that this sentence was misleading and have removed the sentence. Further interpretations of the MT findings are outlined in the response to the reviewer’s next comment below.

8) Lines 396-406: There were significant changes in the deltaMT, but no significant changes in the MT. Any discussion on that? Does that mean future studies should use deltaMT instead of MT to analyze the muscle changes?

We thank this reviewer for highlighting the critical findings of our study. The paragraph beginning on line 369 discusses the results for resting MT and why we suspect these findings occurred. Specifically, the elevation change in this race was low compared to other documented ultramarathons, which may result in less disruption to the cell membrane that would cause swelling in the resting state. 

Line 369: “While prolonged low-load exercise protocols contribute to muscular swelling or edema [28], this was not observed as greater resting rectus femoris MT following the 50km in our study. Elevation changes, especially elevation loss, can increase muscle swelling and inflammation due to the repeat eccentric contractions that occur during a race with large elevation loss [29]. Common to ultramarathon running, many races exceed 3,000 m [3,11,22,30–32] in elevation change over the course of the race. However, by comparison, the current study’s race had a total elevation change just under 762 m with no net gain or loss.”

The next paragraph beginning on line 384 discusses the ∆ MT findings and potential reasons for these findings with a focus on the contracted state. 

Line 384: “When compared with baseline, we observed a significantly higher absolute and % ∆ MT at 24hrs post-race. To our knowledge, this is the first account of MT being reported as the ∆ MT from resting to contracted conditions in this non-elite ultrarunning population. The 24-hr post-race increase in ∆ MT is consistent with other reports of delayed onset of muscle soreness [29,35] and is supported by the clinically meaningful increase in CK at 24hrs post-race.”…..

Line 398: “…The inflammatory response to exercise is highly variable and appears dependent on the extent of the muscle impairment [29]. In our observation, inflammation exhibited post-race may contribute to altered skeletal muscle recruitment patterns at 24-hrs post-race such that over-recruitment [36], as evidenced by greater contracted MT, is required to produce the same force when asked to maximally contract the quadricep [16,36]. This would exhibit as greater ∆ MT 24-hrs post-race. Future studies including EMG while assessing MT in the contracted state are needed to confirm this hypothesis. Alternatively, while our results suggest a potential temporal role of post-race inflammation on elevations in MT 24-hrs post-race, we cannot exclude other explanations. For instance, other studies have documented associations between elevated intramuscular fluid pressure and delayed onset of muscle soreness following repeated eccentric activity [19]. This may be more evident in the muscle during the contracted state, causing the increases in ΔMT observed 24-hrs post-race.”

Collectively, the above-mentioned paragraphs discuss the proposed mechanisms behind each measure, provide potential reasons for why significant differences may not be present in ∆ MT but not the resting state, and advocate for the inclusion of both resting and contracted MT measures as they provide insight into different mechanisms that can be explored using ultrasound (see line 407 as pasted below). 

Line 410: “While the exact mechanisms contributing to changes in ΔMT are not well understood, our findings substantiate the inclusion of MT in both the resting and contracted state to allow for greater insight into changes in muscle morphology in response to acute exercise.”

9) Lines 430&431: “The specific role of calprotectin in response to exercise is not completely understood”

Then why did the authors choose to measure calprotectin? Needs to be explained in the introduction.

Calprotectin was selected due to previous reports of it being release by and subsequently acting on skeletal muscle in response to prolonged contraction and its role in the activation and breakdown of damaged cells through inflammatory signaling pathways (Mortensen et al., J Physiol 2008). This is described in line 81-82 of the introduction and the combined role of IL-6 and calprotectin as it relates to our findings in described in more detail in the revised discussion.

Line 81: Other inflammatory factors such as tumor necrosis factor (TNF)-� [3,9–11], and calprotectin [12,13] are elevated after prolonged running and are involved in the activation and breakdown of damaged cells, including skeletal muscle cells, through inflammatory signaling pathways [12,14]. Together, these factors may contribute to declines in muscle function (e.g., acute decreases in strength) [15] seen after an ultramarathon event and play a role in subsequent recovery [16].

Line 422: “…IL-6 and muscle contractions also promote the release of calprotectin from skeletal muscle [38] which was observed in the present study during and following the 50km race. The specific role of calprotectin in response to exercise is not completely understood, but it has been found to activate cellular pathways promoting inflammation [13,38]. Inflammation induced by elevated concentrations of IL-6 and calprotectin post-race may have a collective effect on motor recruitment patterns [7] or contraction-induced swelling [38] that exhibit as physiological phenotype (i.e. greater absolute and % ∆ MT) 24 hrs post-race.

10) Lines 422-429: The discussion on the role of IL-6 here is very confusing. The greater ΔMT was observed at 24hr post-race when the “IL-6 levels were nearly returned back to baseline”. How does “the greater absolute and % ΔMT observed at 24 hrs post-race” support “IL-6 may aid in muscle regeneration in response to exercise through the classical IL-6 signaling pathway associated with anti-inflammatory properties [6].”? How does this suggest “a delayed inflammatory response”?

As with many cytokines, the peak in plasma concentration often presents earlier than the physiological phenotype (i.e . ΔMT). We recognize that the term “delayed inflammatory response” may not best represent this point and have revised this sentence accordingly. Further, we have rearranged this paragraph to explain the potential role of IL-6 and calprotectin in ΔMT more clearly.

Line 426: “Inflammation induced by elevated concentrations of IL-6 and calprotectin post-race may have a collective effect on motor recruitment patterns [7] or contraction-induced swelling [38] that exhibit as physiological phenotype (i.e. greater absolute and % ∆ MT) 24 hrs post-race”

11) Line 424: Why sIL-6R levels were not affected? Any discussion on that?

The absence of change in sIL-6R suggests that IL-6 was acting on its membrane-bound receptor (as opposed to the soluble receptor). This suggests that IL-6 is playing a role in metabolic recovery post-exercise. This has been further clarified in the discussion. 

Line 416: In the present study we found that plasma IL-6 levels increased substantially throughout the 50km while sIL-6 receptor levels were not significantly affected. Although still slightly elevated, IL-6 levels were nearly back to baseline by 24hrs post-race. IL-6 is consistently elevated in response to exercise even in the absence of muscle damage [38], while elevations in sIL-6R are variable [7]. No changes in sIL-6R suggest that IL-6 is acting primarily through its membrane-bound receptor [8]. Through this pathway, IL-6 most likely aids in fat oxidation and glucose disposal to promote recovery post-exercise [38].

12) Lines 444-446: “The decline in muscle strength… may contribute to …observed increase in ΔMT…”. So, the changes in ΔMT were due to the muscle strength decline? Very confusing.

We have revised this paragraph to better express the role that the strength measures play in the interpretation of our findings.

Line 441: “Both central and peripheral fatigue can explain decrements in force production following an ultramarathon [16,41] and have been shown to persist for well over 30 minutes post-exercise in endurance events lasting several hours [36]. While not directly assessed in this study, these past findings suggest that it is likely that both central (motor unit recruitment) and peripheral fatigue (metabolite buildup and/or impaired excitation-contraction coupling) contribute to reduced post-race knee extensor torque and may also be influenced by inflammation. Specifically, inflammatory factors, including IL-6 [7], can inhibit neural drive by interfering with afferent feedback from exercising muscles [16,42]. In our study, this may be exhibiting as 24 hr post-race declines in muscle strength and compensatory increases in ΔMT representing altered motor unit recruitment [7,16,42].”

13) Line 509: “…delayed onset of muscle inflammation…”. What does this mean? Does it mean the muscle inflammatory response only occurred at a later timepoint like 24hr post-race in this study? Needs to be careful with “delayed onset” here because the acute inflammatory response obviously already initiated quickly post-race or even during the race as indicated by the increased IL-6 levels at 10km and post-race.

This sentence has been revised to better represent the conclusions:

Line 519: “Collectively, these findings indicate the presence of some residual strength impairments and morphological changes to the rectus femoris 24-hrs following a 50km ultramarathon in recreational runners.”

14) One of the main rationales of this study was to investigate the “trajectory of the inflammatory process in non-elite athletes..”(Line 103). It would be nice if the authors discuss whether their findings matched with those of studies in elite athletes.

Aside from training status, several factors are involved in the release of inflammatory cytokines including duration, intensity, and elevation changes within the course. As we do not have a group of elite runners performing the same race, these various confounding variables prevent us from comparing our results to those of elite athletes in other studies. We have added this as a limitation to our discussion.

Line 464: “Findings from this study provide insight into the muscle inflammatory response to ultramarathon running in recreational runners. Differences in race duration, intensity, and elevation change do not allow for direct comparisons between our findings and those from elite runners. Future studies should include both recreational and elite runners to determine whether findings are consistent in individuals with greater fitness levels and training.”

15) It was not clear how the findings from this study can help “develop training plans for longer events and design recovery strategies to optimize skeletal…”(Line 513). Pleaser provide more direct connection between them.

These sentences have been revised:

Line 523: “It not uncommon for recreational ultramarathon runners to participate in stage races, back-to-back long runs, or weekly marathon and 50k distances [2]. Thus, these findings may be used by athletes, coaches, and trainers to optimize recovery strategies for skeletal muscle health, function, and inflammation when partaking in subsequent prolonged running events.”

---

## [Decision Letter · Decision Letter 2]

10 Aug 2022

Muscle thickness and inflammation during a 50km ultramarathon in recreational runners

PONE-D-22-07751R2

Dear Dr. Landers-Ramos,

We’re pleased to inform you that your manuscript has been judged scientifically suitable for publication and will be formally accepted for publication once it meets all outstanding technical requirements.

Kind regards,

Jeremy P Loenneke

Academic Editor

PLOS ONE

Additional Editor Comments (optional):

Reviewers' comments:

Reviewer's Responses to Questions

**Comments to the Author**

1. If the authors have adequately addressed your comments raised in a previous round of review and you feel that this manuscript is now acceptable for publication, you may indicate that here to bypass the “Comments to the Author” section, enter your conflict of interest statement in the “Confidential to Editor” section, and submit your "Accept" recommendation.

Reviewer #1: All comments have been addressed

2. Is the manuscript technically sound, and do the data support the conclusions?

Reviewer #1: Yes

3. Has the statistical analysis been performed appropriately and rigorously? 

Reviewer #1: Yes

4. Have the authors made all data underlying the findings in their manuscript fully available?

Reviewer #1: Yes

5. Is the manuscript presented in an intelligible fashion and written in standard English?

Reviewer #1: Yes

6. Review Comments to the Author

Reviewer #1: Thank the authors for addressing all my major concerns. In this reviewer's opinion, the manuscript now reaches the publication standard.

7. PLOS authors have the option to publish the peer review history of their article (what does this mean?). If published, this will include your full peer review and any attached files.

Reviewer #1: No

---

## [Editor Report · Acceptance letter]

12 Aug 2022

PONE-D-22-07751R2 

Muscle thickness and inflammation during a 50km ultramarathon in recreational runners 

Dear Dr. Landers-Ramos:

I'm pleased to inform you that your manuscript has been deemed suitable for publication in PLOS ONE. Congratulations! Your manuscript is now with our production department. 

Kind regards, 

on behalf of

Dr. Jeremy P Loenneke 

Academic Editor

PLOS ONE